# Closed Boundary Learning for NLP Classification Tasks with the Universum Class

## Abstract

The Universum class, often known as the *other* class or the *miscellaneous* class, is defined as a collection of samples that do not belong to any class of interest. It is a typical class that exists in many classification-based tasks in natural language processing (NLP), such as relation extraction, named entity recognition, sentiment analysis, etc. During data labeling, a significant number of samples are annotated as Universum because there are always some samples that exist in the dataset but do not belong to preset target classes and are not of interest in the task. The Universum class exhibits very different properties, namely **heterogeneity** and **lack of representativeness in training data**; however, existing methods often treat the Universum class equally with the classes of interest. Although the Universum class only contains uninterested samples, improper treatment will result in the misclassification of samples of interest. In this work, we propose a closed boundary learning method that treats the Universum class and classes of interest differently. We apply closed decision boundaries to classes of interest and designate the area outside all closed boundaries in the feature space as the space of the Universum class. Specifically, we formulate the closed boundaries as arbitrary shapes, propose a strategy to estimate the probability of the Universum class according to its unique property rather than the within-class sample distribution, and propose a boundary learning loss to learn decision boundaries based on the balance of misclassified samples inside and outside the boundary. By conforming to the natural properties of the Universum class, our method improves both accuracy and robustness of classification models. We evaluate our method on 6 state-of-the-art works in 3 different tasks, and the F1 score/accuracy of all 6 works is improved. Experimental results also indicate that our method has significantly enhanced the robustness of the model, with the largest absolute F1 score improvement of over $8\%$ on the robustness evaluation dataset. Our code will be released on GitHub.

## 1 Introduction

In classification-based tasks of NLP, quite often we encounter a class named as *other* class, *miscellaneous* class, *neutral* class or *outside (O)* class. Such a class is a collection of samples that do not belong to any class of interest, such as samples of *no relation* class in relation extraction task. We adopt the terminology in (Weston et al., 2006) to designate all such classes as the *Universum class* (U). Universum class exits in various classification-based problems in NLP, such as relation extraction (RE) (Zhang et al., 2017), named entity recognition (NER) (Tjong Kim Sang & De Meulder, 2003), sentiment analysis (SA) (Tjong Kim Sang & De Meulder, 2003), and natural language inference (NLI) (Bowman et al., 2015). To distinguish the Universum class and the rest of the classes, we call the classes of interest as *target classes* (T). The set of all classes (A) in training and testing data can be expressed as $A = U \cup T$

- *Universum class*: A collection of samples that do not belong to any class of interest.
- *Target class*: A class of interest in the task, i.e., one of the classes other than the Universum class.

The sample compositions of the Universum class and target classes are usually very different. Figure 1(a) provides some samples of a target class (*entity-destination*) and the Universum class (*other*) in

relation extraction. From the examples, we can observe that the *entity-destination* samples adhere to an intrinsic pattern: an entity goes somewhere. However, the three examples of the *other* relation type are vastly dissimilar and do not exhibit any common intrinsic pattern. In fact, the Universum samples are labeled according to an extrinsic pattern: they do not belong to any of the predefined target classes.

We further highlight the differences between the Universum class and target classes in two properties. (1) **Heterogeneity**: The universum class is composed of heterogeneous samples, which may form multiple clusters in the feature space of the test set, as illustrated by the green samples in Figure 1(b). This is because the Universum class, as the class name "other" implies, contains all potential implicit classes that are not explicitly defined in the task. For example, in the *other* samples given in Figure 1(a), implicit classes may include the entity-parallel relationship, the entity-fill relationship, and the entity-narrative relationship. Although such heterogeneous samples are easily mapped into a compact cluster for the training set, it is problematic for the test set. Since the natural predictive rule of the Universum class is an *extrinsic* pattern that does not belong to any target classes, the model is more likely to fit the noise in the Universum class by memorizing various peculiarities of *intrinsic* heterogeneous samples rather than finding the general predictive rule. Considering the data distribution of the test set differs from the training set, only memorizing various peculiarities can easily lead to overfitting and result in an accuracy drop. Moreover, the lack of robustness is another consequence of inability to find the extrinsic predictive rule for the Universum class.
(2) **Lack of Representativeness in Training Data**: The Universum class is the complementary set of predefined target classes in the task. Therefore, it contains all possible implicit classes, i.e., classes not explicitly defined in the task but may appear in the real world. In this case, Universum samples in the training data are unable to sufficiently represent all possible patterns of the genuine distribution of the Universum class. As depicted in Figure 1(b), gray samples represent Universum samples in the test set that are not represented by the training data. Classifiers with open boundaries are prone to misclassifying unseen samples in the test set that is not represented by the training data.

Despite the substantial difference between the target classes and the Universum class, this issue has long been neglected by the NLP research community. The majority of works (Zhu & Li, 2022; Ye et al., 2022; Wan et al., 2022; Pouran Ben Veyseh et al., 2022; Tian et al., 2021; Fu et al., 2021; Li et al., 2021b) treat the Universum class and target classes equally. Typically, a linear layer and a softmax function are applied at the end of the model to generate open decision boundaries, which we believe are inappropriate for tasks containing the Universum class.

How to take into account the different properties of the Universum class and target classes in classifier design? In this work, we propose a closed boundary learning method for classification-based tasks with the Universum class. Traditional methods often adopt open boundary classifiers since the problem is under the closed-world assumption. However, the open decision boundaries can easily misclassify Universum samples, as illustrated in Figure 1(c). Therefore, we propose to use closed boundary classifiers as shown in Figure 1(d). We constrain the space of target classes to be closed spaces and designate the area outside all closed boundaries in the feature space as the space of the Universum class. This treatment perfectly fits the nature of the Universum class: a sample is marked as the Universum if it does not belong to any target class during labeling. The aforementioned two properties are also well addressed in this way.

The main contributions of this work are summarized as follows:

- We bring attention to an important issue that is frequently neglected in classification-based NLP tasks like NER, RE, SA, and NLI: the Universum class, such as the *other* class, exhibits very different properties from target classes and should be treated differently.

- Methodologically, we generate closed boundaries with arbitrary shape, which include the commonly used spherical-shaped boundary as a special case. In addition, we leverage the information of both target classes and the Universum class to learn the decision boundary. We propose a boundary learning loss to generate the boundary based on the balance of misclassified Universum samples and the target class samples. In contrast to the intuitive two-step pipeline method, where Universum identification and multi-class classification may lead to error propagation, we develop a strategy to estimate the probability of the Universum class without relying on its intrinsic sample distribution and learn the classification of the Universum class and target classes jointly.

| *Entity-Destination* Relation | *Other* Relation |
|---|---|
| The famous **actress** arrived at the **airport**. | The **captain** and **crews** are grateful for the support. |
| Quake **survivors** moved into makeshift **houses**. | The **room** was filled with huge **canvases**. |
| The research **team** is going into the deep **jungle**. | The **stories** are narrated through **dance**. |
| ... | ... |

(a)

(b)  (c)  (d)

Figure 1: Illustration of distinction between the Universum class. (a) Samples selected from the SemEval 2010 Task 8 dataset on relation extraction. (b) The distribution of target classes (class 1, 2, 3) and the Universum class (class 4). In particular, the gray samples represent Universum samples in the test set that are not represented by the training data. (c) The open decision boundaries obtained by traditional classifiers. (d) The arbitrary closed boundaries obtained by our proposed method.

- We apply our closed boundary learning framework to 6 state-of-the-art (SOTA) works in 3 different tasks with Universum class. In all 6 experiments, the accuracy or F1-score is improved over the original work. Experimental results further show that the enhanced accuracy is attributed to the improvement of both the Universe class and the target class.

- By conforming to the natural properties of the Universum class, our method provides a more reasonable way of learning and enhance the robustnessof the model as well. Experimental results indicate that the model's robustness has been significantly enhanced by our method, with a largest absolute F1 score improvement over $8\%$ on the robustness evaluation dataset.

## 2 RELATED WORKS

### 2.1 CLASSIFICATION TASKS WITH THE UNIVERSUM CLASS

The Universum class widely exists in classification based tasks in NLP, such as relation extraction (Zhang et al., 2017), named entity recognition (Tjong Kim Sang & De Meulder, 2003), and sentiment analysis (Jiang et al., 2019), as summarized in Table 1. It should be noted that the span-based methods (Zhu & Li, 2022; Li et al., 2021a) enumerate all possible spans and classify the entity from the whole set of spans, which introduces an extra *other* class. Despite the heterogeneity and lack of representativeness of the Universum class, current works (Zhu & Li, 2022; Wan et al., 2022; Fu et al., 2021; Tian et al., 2021; Chen et al., 2021; Li et al., 2021b; 2020; Yu et al., 2020) solve the classification problems containing the Universum class as normal multi-class classification problems and treat the Universum class and target classes equally.

### 2.2 CLOSED BOUNDARY LEARNING METHODS

Closed boundaries are often adopted in research fields of out-of-distribution (OOD) detection (Gomes et al., 2022; Ren et al., 2021; Chen et al., 2020), open set recognition (Zhang et al., 2021; Liu et al., 2020), anomaly detection (Zong et al., 2018), and outlier detection (Sharan et al., 2018; Sugiyama & Borgwardt, 2013). We adopt the terminology of generalized OOD detection (Yang et al., 2021b) to encompass these problems and analyze the differences.

### 2.2.1 Difference in Problem Setting

The problem we raised in this paper lies in **the intersection of open-world and closed-world assumptions.** Classification tasks can be categorized into problems based on closed-world assumption and open-world assumption (Yang et al., 2021b). The classification problem with the Universum class shares similarities and differences with both assumptions. The tasks highlighted in this paper are treated under the closed-world assumption. Nevertheless, the Universum samples in these tasks are analogous to OOD samples in open-world assumption. On the other hand, the problem we raised differs significantly from open-world problems. This is because the OOD samples are not available in the training data in generalized OOD detection problems, whereas a considerable number of Universum samples are included in the training data in our problem setting. The information of existing Universum samples is important to generate accurate decision boundaries in our problem.

### 2.2.2 Difference in Methodology

By definition, the OOD detection problem assumes that the training data do not contain any OOD samples. However, a branch of the OOD studies, known as outlier exposure (Katz-Samuels et al., 2022; Ming et al., 2022; Yang et al., 2021a; Thulasidasan et al., 2021; Mohseni et al., 2020; Hendrycks et al., 2018), introduces auxiliary outlier data during training. The introduced auxiliary data makes it close to the format of our raised classification problems with the Universum class. However, outlier exposure methods are not suitable for our problem. The outlier exposure method mostly adopts a two-step approach that consists of multi-class classification and OOD identification. The OOD identification step distinguishes OOD and ID samples based on a score obtained by cross entropy (Yang et al., 2021a; Thulasidasan et al., 2021; Mohseni et al., 2020; Hendrycks et al., 2018), and energy function (Katz-Samuels et al., 2022; Ming et al., 2022). However, both cross entropy and energy function are monotonically varying. As a result, the decision boundary derived from a threshold score of the monotonically varying function is an open boundary, which leaves the heterogeneity and representativeness issues we pointed out in this paper still unresolved. In addition, the two-step approach will also suffer from error propagation.

From a methodological point of view, our work is also different from the works in generalized OOD using closed boundaries. In generalized OOD studies, the closed boundaries are formulated by the classic density-based method (Pidhorskyi et al., 2018; Hu et al., 2018), one-class classification method (Reiss et al., 2021; Ruff et al., 2018), or distance-based method (Gomes et al., 2022; Sun et al., 2022; Zhang et al., 2021; Zaeemzadeh et al., 2021; Shu et al., 2020). The distance-based methods are limited to spherical boundary shapes but our method can generate arbitrary shape boundaries. The one-class classification method formulates only one closed boundary between positive and negative samples while our work generates closed boundaries for every target class. Finally, only positive samples are used to learn decision boundaries in density-based method, while both target class samples and Universum samples are used in our work.

Table 1: The tasks and datasets that the Universum class exists.

| Task | Dataset | Label Name | Proportion |
|------|---------|------------|------------|
| Relation Extraction | SemEval 2010 Task 8 (Hendrickx et al., 2019) | Other | 17.4% |
| Relation Extraction | TARCED (Zhang et al., 2017) | No relation | 79.5% |
| Named Entity Recognition | CoNLL-2003 (Tjong Kim Sang & De Meulder, 2003) | Miscellaneous | 14.6% |
| Named Entity Recognition (span based method) | CoNLL-2003 (Tjong Kim Sang & De Meulder, 2003) | Other | >90% |
| Aspect Category Sentiment Analysis | MAMS (Jiang et al., 2019) | Neutral | 43.4% |

## 3 Method

**Problem Definition**: The goal of our proposed method is to learn closed decision boundaries for target classes and jointly classify the Universum samples and target samples. In order to make our proposed method compatible with most existing classification methods, the starting point of our method is the representations of the final layer of classification models, which is a linear layer that maps data from high-dimensional feature space to a lower-dimension space. We denote the sample representations of the final linear layer as $\mathbf{H} = \{\mathbf{h}_0, \mathbf{h}_1, \ldots, \mathbf{h}_{N-1}\} \in \mathbb{R}^{N \times l}$, where $N$ is the number of samples, and $l$ is the output dimension of the linear layer.

## 3.1 DEFINING THE UNIVERSUM CLASS

Universum class exists in many tasks and datasets as we summarized in Table 1. Notably, the Universum class has various names such as *other* and *miscellaneous*, etc. In sentiment analysis, the *neutral* class can be considered as the Universum class because the word *neutral* is defined as "having no strongly marked or positive characteristics or features", which means the *neutral* class is a collection of all samples without strong emotions. Similarly, the *no relation* class in the relation extraction task can be considered as the Universum class.

## 3.2 PRETRAINING

Our method estimates the probability distribution of target classes based on their sample distributions. In order to avoid estimation based on randomly initialized weight and speed up the learning process, we employ N-pair loss (Sohn, 2016) for pretraining, making sample representations be of small intra-class distance and large inter-class distance. Notably, in accordance with the nature of the Universum class, we make a change that does not require the model to reduce the intra-class distance of Universum samples during the pretraining.

## 3.3 GENERATING CLOSED BOUNDARY OF ARBITRARY SHAPE FOR TARGET CLASSES

Existing closed boundary classification methods mainly use spherical shape boundaries (Zhang et al., 2021; Liu et al., 2020); however, we argue that the spherical shape may not be the optimal solution because data samples are unlikely to perfectly fit into a sphere, and a spherical shape boundary may produce misclassifications. We adopt the Gaussian mixture model (GMM) and the threshold value to generate boundaries with arbitrary shapes.

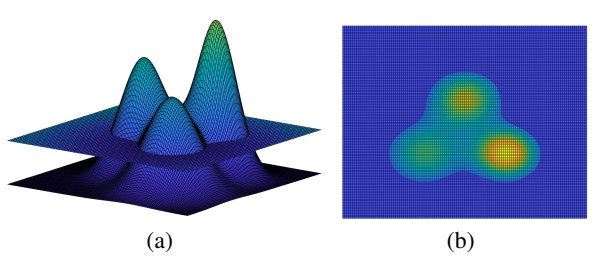

(a)        (b)

Figure 2: Illustration of generating arbitrary shape boundaries.

### 3.3.1 GAUSSIAN MIXTURE MODEL

We apply GMM with $m$ components to estimate the class conditional probability distribution for each target class $C_i$, and further derive the joint probability estimation for each class.

$$p(\mathbf{h}_k \mid C_i) = \sum_{i=1}^{m} \pi_i \mathcal{N}\left(\mathbf{h}_k; \boldsymbol{\mu}_i, \boldsymbol{\Sigma}_i\right) \tag{1}$$

$$p(\mathbf{h}_k, C_i) = p(\mathbf{h}_k \mid C_i)p(C_i) \tag{2}$$

where $\mathbf{h}_k$ denotes the input feature vector of the $k$th sample, $\boldsymbol{\mu}_i$ and $\boldsymbol{\Sigma}_i$ are the estimated mean vector and covariance matrix of the $i$th Gaussian components, respectively. $\pi_{ij}$ is the non-negative mixture weight under the constraint that $\sum_{j=1}^{m} \pi_{ij} = 1$. $\boldsymbol{\mu}_i$, $\boldsymbol{\Sigma}_i$, and $\pi_{ij}$ are all learnable parameters in the model.

It should be noted that the accuracy of GMM estimation is positively related to the number of samples used (Psutka & Psutka, 2019), which should be at least equal to the dimension of the data. Therefore, our method is not suitable for zero-shot or few-shot settings. Nevertheless, given that the dimension $l$ in our method is very small and the initialized GMM parameters will be fine-tuned by the neural network, constraints on sample size can be easily overcome on the most classification tasks.

According to Bayes Theorem, the posterior probability $p(C_i \mid \mathbf{h}_k) = \frac{p(\mathbf{h}_k \mid C_i)p(C_i)}{p(\mathbf{h}_k)}$. Since we are interested in $\mathrm{argmax}_{C_i} \frac{p(\mathbf{h}_k \mid C_i)p(C_i)}{p(\mathbf{h}_k)}$, the decision can be made based on joint probability $p(\mathbf{h}_k, C_i)$.

### 3.3.2 ARBITRARY SHAPE BOUNDARY

**Geometrical View**: Inspired by the DENCLUE algorithm in generating arbitrary shape clusters (Hinneburg & Keim, 1998), we introduce a threshold value $\xi_i$ for each target class. A closed boundary

of arbitrary shape is formulated by points satisfying $p(\mathbf{h}, C_i) = \xi_i$. Figure 2 is an illustration of formulating an arbitrary shape boundary in a three-dimensional space. A sample is assigned to class $C_i$ if it is located inside the closed boundary. If the number of components of the GMM is set to one, then the shape of the boundary becomes spherical, with its center and covariance matrix specified by $\boldsymbol{\mu}_0$ and $\boldsymbol{\Sigma}_0$, respectively. In this sense, the commonly used spherical shape boundary (Zhang et al., 2021; Liu et al., 2020) is a special case of our method.

Notably, the threshold values $\Xi = \xi_1, \xi_2, \cdots, \xi_{n-1}$ are a learnable parameters, which eliminates the laborious process of hyperparameter tunning. Specifically, it is learned based on the balance of misclassified samples inside and outside the boundary through our proposed boundary learning loss, which is introduced later.

**Probabilistic View**: The above geometrical process can be described as:

$$\begin{cases} \mathbf{h}_k \in C_i \text{ if } p(\mathbf{h}_k, C_i) > \xi_i \\ \mathbf{h}_k \notin C_i \text{ if } p(\mathbf{h}_k, C_i) \leq \xi_i \end{cases} \tag{3}$$

### 3.4 EXTRINSIC RULE-BASED PROBABILITY ESTIMATION FOR THE UNIVERSUM CLASS

Our method learns the Universum class and target classes jointly to prevent error propagation by intuitively classifying target classes and the Universum class in two steps. However, unlike target classes that estimated by intrinsic sample distribution, estimating the probability distribution of the Universum class must be based on an extrinsic rule that does not belong to any target classes. We propose an extrinsic rule-based probability estimation method to address this issue.

#### 3.4.1 MOTIVATION AND THE ESTIMATION

We classify the samples of Universum class and target classes based on the following rules in this work:

- *Rule 1*: A sample is assigned to the Universum class if it is not located inside any of the closed boundaries of target classes.
- *Rule 2*: A sample is assigned to the target class with the highest $p(\mathbf{h_k}, C_i)$ if it is located inside at least one closed boundary.

An intuitive way to incorporate the above rules is a two-step method consists of Universum class detection and target classes classification. However, such a pipeline method has the issue of error propagation. Another way is to use the distance relationship between the sample point and centers of target classes. However, the distance, e.g., Mahalanobis distance, is only a simple special case obtained from Section 3.4. In addition, the distance-based method is incompatible with cross-entropy loss, which has the benefit of being directly related to accuracy. Finally, general probability estimation methods, such as the GMM model, exploit intrinsic sample distributions, which fail to overcome the special properties of the Universum class and do not conform to Rule 1. Therefore, a strategy need to be devised to convert Rule 1 into a probability expression, while simultaneously facilitates the learning of the neural network.

For compliance with Rule 1, the estimated probability of the Universum class must satisfy the following two conditions: for Universum class samples: $\forall i : p(\mathbf{h}_k, U) > p(\mathbf{h}_k, C_i)$ and for target class samples: $\exists i : p(\mathbf{h}_k, U) < p(\mathbf{h}_k, C_i)$. We can leverage the relationship between $\xi_i$ and $p(\mathbf{h}_k, C_i)$ defined in Equation 3 to construct the estimation of $p(\mathbf{h}_k, U)$ that satisfies the above two conditions. In addition, to enhance the learning of neural networks, the gradient obtained from an Universum sample should move this sample away from its closest target class boundary. Therefore, we also involve $\max(p(\mathbf{h}_k, C_i))$, the probability of the closest target class of a Universum sample, to guide the Universum sample move away from target class boundaries. We propose to estimate the probability distribution of the Universum class as follows:

$$p(\mathbf{h}_k, U) = \lambda \frac{\xi_u^2}{p(\mathbf{h}_k, C_u)} + (1 - \lambda) \frac{\xi_v^2}{p(\mathbf{h}_k, C_v)} \tag{4}$$

$$\lambda = \begin{cases} 1, p(\mathbf{h}_k, C_u) > \xi_u \\ 0, p(\mathbf{h}_k, C_u) \leq \xi_u \end{cases}, \begin{cases} u = \operatorname{argmax}_i \frac{p(\mathbf{h}_k, C_i)}{\xi_i} \\ v = \operatorname{argmax}_i p(\mathbf{h}_k, C_i) \end{cases} \tag{5}$$

$\xi_i$ is the threshold value of target class i, and $u, v \in \{1, 2, \ldots, n-1\}$.

### 3.4.2   ANALYSIS OF THE PROPOSED ESTIMATION

For estimated probability of the Universum class in Equation 4, two cases are possible.
**Case 1**: $p(\mathbf{h}_k, C_u) > \xi_u$, i.e., sample $\mathbf{h}_k$ is located inside at least one closed boundary.
In this case, we have

$$p(\mathbf{h}_k, U) = \xi_u \frac{\xi_u}{p(\mathbf{h}_k, C_u)} < \xi_u < p(\mathbf{h}_k, C_u)$$

Since $p(\mathbf{h}_k, U) < p(\mathbf{h}_k, C_u)$, the model will select the target class $i$ with the highest $p(\mathbf{h}_k, C_i)$, which fits perfectly with Rule 2.
**Case 2**: $p(\mathbf{h}_k, C_u) \leq \xi_u$, i.e., the sample $\mathbf{h}_k$ distribute outside every closed boundary.
Combining the condition of case 2 and Equation 5, we have

$$\forall i \in \{1, 2, \ldots, n-1\} : \frac{p(\mathbf{h}_k, C_i)}{\xi_i} \leq \frac{p(\mathbf{h}_k, C_u)}{\xi_u} \leq 1$$

$$\text{i.e.,} \forall i \in \{1, 2, \ldots, n-1\} : p(\mathbf{h}_k, C_i) \leq \xi_i \tag{6}$$

Combining Equation 5 and Equation 6, we can derive that:

$$\forall i \in \{1, 2, \ldots, n-1\} : p(\mathbf{h}_k, U) = \xi_v \frac{\xi_v}{p(\mathbf{h}_k, C_v)} \geq \xi_v \geq p(\mathbf{h}_k, C_v) \geq p(\mathbf{h}_k, C_i) \tag{7}$$

In case 2, from Equation 6 and Equation 3, we can learn that sample $\mathbf{h}_k$ is located outside all closed boundaries of target classes. In this case, the probability of Universum class $p(\mathbf{h}_k, U)$ obtains the largest value, as shown in Equation 7. Therefore, Rule 1 is perfectly expressed by the proposed probability estimation of the Universum class.

## 3.5   BOUNDARY LEARNING LOSS

To facilitate the learning of the closed decision boundaries, we propose a boundary learning loss below. The decision boundary should be adjusted to the balance of misclassified samples inside and outside the boundary. For example, if samples of class $j$ distribute inside the boundary of class $i$, then the boundary should contract to exclude such samples and vice versa.

$$L_{bl} = \frac{1}{M} \sum_{i=1}^{n-1} \left( \sum_{k \in \mathbb{O}} w_k \log \frac{\xi_i}{p(\mathbf{h}_k, C_i)} + \sum_{l \in \mathbb{I}} w_l \log \frac{p(\mathbf{h}_l, C_i)}{\xi_i} \right)$$

$M$ is the total number of misclassified samples for all boundaries, $n$ is the number of classes, $\mathbb{O}$ and $\mathbb{I}$ denote the set of training samples misclassified outside and inside the decision boundary $i$, respectively. The weights in the loss function are $w_k = \frac{p(\mathbf{h}_k, C_i)}{p(\mathbf{h}_k, C_i) + \xi_i}$, $w_l = \frac{\xi_i}{p(\mathbf{h}_l, C_i) + \xi_i}$, and they are detached and cut off the gradient. Weights $w_k$ and $w_l$ have smaller values for samples located far from the boundary, enabling the boundary to be adjusted primarily on the basis of easily and semi-hard negatives instead of hard negatives.
During training, we sum the cross-entropy loss and boundary learning loss for optimization. The boundary learning loss forces misclassified samples to be distributed in the correct region, which works well with cross-entropy loss. Moreover, the boundary learning loss is defined as the sum of each boundary's the false positive (FP) samples and false negative (FN) samples. For instance, if there are considerably more FN samples than FP samples, the loss caused by FN samples will be larger, causing the boundary to adjust to eliminate more FN samples and balance the number of FN and FP samples. Hence, the boundary learning loss can further improve the F1 score.

## 4   EXPERIMENTS

### 4.1   DATASETS AND BASELINES

Our proposed method is evaluated on six different SOTA models on three classic datasets in NLP, including SemEval 2010 Task 8 (Hendrickx et al., 2019), MAMS (Jiang et al., 2019), and CoNLL-2003 (Tjong Kim Sang & De Meulder, 2003). The proportion of Universum samples in the

Table 2: The overall performance of applying closed boundary learning on baseline models.

| Task | Method | F1/accuracy | p-value |
|---|---|---|---|
| NER | SpanNER (Fu et al., 2021) | 92.09±0.16 | |
| | SpanNER (Fu et al., 2021) + OECC | 91.22 ± 0.12 | < 0.001 |
| | SpanNER (Fu et al., 2021) + COOLU | **93.50**±0.13 | |
| | BS (Zhu & Li, 2022) | 92.53±0.02 | |
| | BS (Zhu & Li, 2022) + OECC | 91.88 ± 0.15 | < 0.01 |
| | BS (Zhu & Li, 2022) + COOLU | **93.17**±0.13 | |
| RE | A-GCN (Tian et al., 2021) | 88.67±0.18 | |
| | A-GCN (Tian et al., 2021) + OECC | 88.00 ± 0.09 | < 0.01 |
| | A-GCN (Tian et al., 2021) + COOLU | **89.33**±0.20 | |
| | TaMM (Chen et al., 2021) | 88.76±0.23 | |
| | TaMM (Chen et al., 2021) + OECC | 88.17 ± 0.18 | < 0.01 |
| | TaMM (Chen et al., 2021) + COOLU | **89.47**±0.21 | |
| ACSA | AC-MIMLLN (Li et al., 2020) | 76.13±0.29% | |
| | AC-MIMLLN (Li et al., 2020) + OECC | 74.02 ± 0.68% | < 0.01 |
| | AC-MIMLLNN (Li et al., 2020) + COOLU | **77.35**±0.42% | |
| | SCAPT (Li et al., 2021b) | 84.13±0.19% | |
| | SCAPT (Li et al., 2021b) + OECC | 83.36 ± 0.27% | < 0.01 |
| | SCAPT (Li et al., 2021b) + COOLU | **85.06**±0.23% | |

three datasets are 17.4% (highest in 19 classes), 90%, and 43.4%. respectively. It is noteworthy that the ratio of Universum class in the NER task is not calculated from the *miscellaneous* samples in the dataset but from the *other* samples which are introduced by the span-based method (Zhu & Li, 2022; Fu et al., 2021).
We evaluate the effectiveness of our proposed **ClO**sed b**O**undary **L**earning for classiciation with the **U**niversum class (**COOLU**) on 6 SOTA works, including SpanNER (Fu et al., 2021), BS (Zhu & Li, 2022), A-GCN (Tian et al., 2021), TaMM (Chen et al., 2021), AC-MIMLLN (Li et al., 2020), and SCAPT (Li et al., 2021b). Moreover, OECC (Papadopoulos et al., 2021), a recent outlier exposure method for OOD, is also applied to these works for comparison with our method. The implementation details are in the Appendix.

## 4.2 OVERALL EXPERIMENTAL RESULTS

Table 2 shows the overall results for all 6 models. Models with our proposed closed boundary learning outperform the original models with open classifiers on NLP tasks containing the Universum class. The overall accuracy or F1 score is improved on all six models we evaluated, with the largest improvement from 92.09 to 93.50 in F1 score. In addition, statistical tests between the accuracy/F1 score of the baseline models and our method. indicate that the improvement brought about by our COOLU method is statistically significant.
Our proposed COOLU method also perform better than OECC (Papadopoulos et al., 2021). To our knowledge, there has been no work special designed for Classifying with the Universum class. OECC as an outlier exposure method, although its problem setting is the closest to ours, is still not suitable. The decrease in accuracy caused by applying OECC is due to: (1) error propagation resulting from outlier detection and multi-class classification in two steps and (2) instability of manually setting thresholds.

## 4.3 A CLOSER LOOK AT THE MICROF1, PRECISION AND RECALL

The improvement of overall performance is not only attributed to the improvement of the Universum class, but also to the improvement of all classes as a whole. We show the micro F1 score of each class by applying closed boundary learning on SpanNER in Table 3. The micro F1 score of *other* class, which is introduced by the span-based method, is not listed in the table because the overall F1 score of the task depends on the F1 score of the four entity types below. The micro F1 score is improved in all classes, with the absolute improvement of 2.43, 0.29, 1.31, and 4.02, respectively.

Table 3: The micro F1 score of SpanNER (Fu et al., 2021) with and without closed boundary learning.

| Method | ORG | | | PER | | | LOC | | | MISC | | |
|---|---|---|---|---|---|---|---|---|---|---|---|---|
| | P | R | F1 | P | R | F1 | P | R | F1 | P | R | F1 |
| SpanNER | 89.92 | 89.81 | 89.87 | 97.74 | 96.47 | 97.11 | 93.05 | 93.88 | 93.46 | 78.99 | 82.83 | 80.87 |
| SpanNER + COOLU | **94.29** | 90.30 | **92.30** | **98.54** | 96.29 | **97.40** | **96.17** | 93.41 | **94.77** | **88.02** | 81.97 | **84.89** |

Table 4: Comparison of model's robustness with and without closed boundary learning.

| | CrossCategory | OOV | SpellingError | AppendIrr |
|---|---|---|---|---|
| SpanNER | 77.06 | 75.14 | 76.09 | 87.34 |
| SpanNER + COOLU | **81.39** | **83.15** | **81.12** | **89.89** |

| | InsertClause | SwapEnt | SpellingError | AppendIrr |
|---|---|---|---|---|
| A-GCN | 77.84 | 86.8 | 77.36 | 86.59 |
| A-GCN + COOLU | **80.17** | **88.69** | **78.71** | **88.22** |

In addition, our proposed closed boundary learning significantly improves the precision score of all classes, with the largest absolute gain being 9.03. We also observe a slight drop in recall as a result of replacing open decision boundaries with closed ones. But the increase in precision considerably outweighs the decrease in recall. The relatively poor performance of the baseline method in precision score further proves our argument that the Universum class is easily misclassified if its special properties are neglected. The improvement in precision score indicates that the proposed closed decision boundaries of target classes can effectively avoid misclassification of the Universum class into target classes. Simultaneously, as fewer Universum samples are misclassified, more Universum samples are correctly identified.

## 4.4 MODEL ROBUSTNESS EVALUATION

What's the relationship between accuracy and robustness? Some studies theoretically identify a trade-off between robustness and accuracy (Tsipras et al., 2019; Zhang et al., 2019; Raghunathan et al., 2020), another finding shows no inherent trade-off (Yang et al., 2020). Regardless, it is challenging to improve both accuracy and robustness of the model, which is achieved by our proposed method.

We evaluate the robustness of the model based on TextFlint (Wang et al., 2021), a robustness evaluation toolkit for NLP tasks. Specifically, TextFlint generate perturbations of the test data by universal transformations and task-specific transformations, and the robustness of the model is evaluated using the transformed test dataset. The terms "CrossCategory", "OOV", "SpellingError", etc. in Table 4 are different ways of transforming the test data. Detail information of these transformation methods are illustrated in Appendix B.3. It can be observed from the Table 4 that the robustness of SpanNER (Fu et al., 2021) improved significantly by applying our proposed COOLU method, with the improvement of absolute F1 score of 4.44, 8.01, 5.03, and 2.55, respectively. In addition, although the improvement of the F1 score in A-GCN (Tian et al., 2021) is less than 1, the robustness of the model is considerably improved. The absolute F1 score on robustness evaluation datasets are improved at 2.33, 1.89, 1.35, and 1.63, respectively. The improvement in robustness validates our argument that our method conform to the natural properties of the Universum class and successfully teach the model the extrinsic predictive rule rather than overfitting to various peculiarities of heterogeneous Universum samples.

## 5 CONCLUSION

In this work, we highlight an overlooked issue in classification-based NLP tasks that the Universum class is treated equally with target classes despite their significant differences. We propose a closed boundary learning method COOLU as a solution , which conforms the natural properties of the Universum class. Specifically, we generate closed boundaries with arbitrary shapes, develop an extrinsic rule-based strategy to estimate the probability of the Universum class, and propose a boundary learning loss to adjust decision boundaries based on the balance of misclassified samples inside and outside the boundary. Experimental results demonstrate that our proposed COOLU method improves the accuracy of all SOTA baseline models and simultaneously enhances model's robustness.

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

## A  COMPACTNESS OF THE UNIVERSUM CLASS OF THE TEST SET

We evaluate the compactness of the Universum class and target classes on the test data and depict the result in Figure 3. The representation of test samples after learning with open boundary classifiers by SpanNER (Fu et al., 2021) is used for evaluation. We evaluate the compactness based on the root-mean-square standard deviation (RMSSD) (Sharma, 1996), and the mean square distance (MSD) (Xie & Beni, 1991), which are commonly used in clustering studies to evaluate compactness of a cluster. The smaller the RMSSD or MSD, the better the compactness. It is illustrated in Figure 3 that the compactness of the "OTHER" class is significantly worse than target classes. Notably, the class with the second-worse compactness is the "MISC" class, i.e., the "miscellaneous" class, which is also a type of the Universum class.

The compactness evaluation results reflect our view that although the heterogeneous samples are easily mapped into a compact cluster for the training data, it is problematic for the test data. Since the natural predictive rule of the Universum class is an *extrinsic* pattern that not belong to any target classes, the model is more likely to fit the noise in the Universum class by memorizing various peculiarities of *intrinsic* heterogeneous samples rather than finding the general predictive rule. In this case, it is hard for the model to map all test samples of the Universum class into the compact space learned from training. The Universum samples that deviate from the original compact space are easily misclassified. On the other hand, target classes do not encounter this problem because the general predictive rule for target classes is intrinsic.

Table 5: The pretrained models chosen for each baseline model and the corresponding F1 score/accuracy reported in the original paper.

| Method | Pretrained Model | Reported F1/accuracy |
|---|---|---|
| SpanNER (Fu et al., 2021) | BERT-base | 92.28 |
| BS (Zhu & Li, 2022) | RoBERTa | 93.65 |
| A-GCN (Tian et al., 2021) | BERT-base | 89.16 |
| TaMM (Chen et al., 2021) | BERT-base | 89.18 |
| AC-MIMLLN (Li et al., 2020) | Glove | 76.42 |
| SCAPT (Li et al., 2021b) | BERT-base | 85.24 |

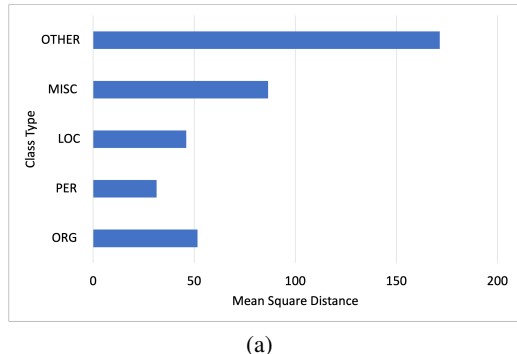
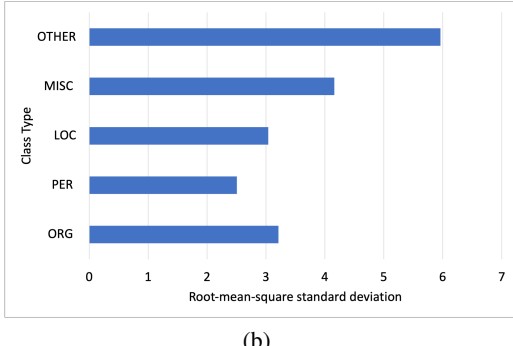

(a)                    (b)

Figure 3: The compactness evaluation of the Universum class and target classes of the test data of NER task.

## B    IMPLEMENTATION DETAILS

### B.1    BASELINE MODELS

We reproduce the baseline models based on the officially released source code, and apply closed boundary learning on the source code. All reported results are the average of three runs. It should be noted that some results of baseline models are slightly different from those given in the original papers due to the variations in random seeds and package versions when reproducing baseline models from their officially released codes. Nevertheless, baseline models and models with closed boundary learning are fairly compared in our work under the same random seed and deep learning environment. In the six baseline models we selected, different results based on multiple language models are often reported in one work. We choose one of the pretrained models used in each work and reproduce the baseline models. The pretrained language model we used in each baseline and their reported results are summarized in Table 5.

### B.2    TRAINING PROCESS

During pretraining process, all parameters of the original model $\boldsymbol{\theta}$ are learned. We employ GMM estimation on training data after pretraining and obtain the initial value of $\boldsymbol{\mu}_i$, $\boldsymbol{\Sigma}_i$, and $\pi_{ij}$, where $i \in \{1, 2, \cdots, n-1\}$, $j \in \{1, 2, \cdots, m\}$. $n$ is the number of classes, and $m$ is the number of GMM components. We typically select $m = 4$ in our experiments. The threshold values $\xi_i$ is initialized around the $a$ quantile of $p(\mathbf{h}_k, C_i)$ values ($k \in \{0, 1, \cdots, N_i - 1\}$), where $a$ is the accuracy or F1 score of the original model. With our extrinsic rule-based probability estimation for the Universum class, we obtain $[p(\mathbf{h}_k, C_1), p(\mathbf{h}_k, C_2), \cdots, p(\mathbf{h}_k, C_{n-1}), p(\mathbf{h}_k, U)]$. Then, the original model parameters $\boldsymbol{\theta}$, GMM parameters $\boldsymbol{\mu}_i$, $\boldsymbol{\Sigma}_i$, $\pi_{ij}$ and threshold values $\xi_i$ are learned by cross-entropy loss and our proposed boundary learning loss.

Table 6: The comparison of our method and the method optimizing F1 score.

| Task | Method | F1 |
|------|--------|-----|
| NER | SpanNER (Fu et al., 2021) | 92.09 |
| | SpanNER (Fu et al., 2021) + ML$^E$ | 92.43 |
| | SpanNER (Fu et al., 2021) + COOLU | **93.50** |
| RE | A-GCN (Tian et al., 2021) | 88.67 |
| | A-GCN (Tian et al., 2021) + ML$^E$ | 88.61 |
| | A-GCN (Tian et al., 2021) + COOLU | **89.33** |
| ACSA | AC-MIMLLN (Li et al., 2020) | 76.13% |
| | AC-MIMLLN (Li et al., 2020) + ML$^E$ | - |
| | AC-MIMLLNN (Li et al., 2020) + COOLU | **77.35**% |

## B.3 ROBUSTNESS EVALUATION

We evaluate the robustness of the model based on TextFlint (Wang et al., 2021), a robustness evaluation toolkit for NLP tasks. There are two kinds of transformations provided by TextFlint to generate the robust evaluation dataset, namely universal transformation and task-specific transformation. We adopt two universal transformations and two task-specific transformations to the test set of NER and RE task and generate four robustness evaluation datasets for each task. The terms of different transformations are explained below.

- "SpellingError": Universal transformation. Brings slight errors to words in the test samples.
- "AppendIrr": Universal transformation. Add irrelevant information to test samples.
- "CrossCategory": Task-specific transformation for NER. Replace the entity spans with substitutions from a different category.
- "OOV": Task-specific transformation for NER. Replace the entity spans with substitutions out of vocabulary.
- "InsertClause": Task-specific transformation for RE. Change sample sentences by appending adjuncts from the aspect of dependency parsing.
- "SwapEnt": Task-specific transformation for RE. Swap the named entities in a sentence into entities of the same type.

Specifically, the models are trained and validated on the original training set and validation set, but the test set is transormed into the robustness evaluation dataset by the transformations proposed by TextFlint. Then, modelw are tested on the transformed test set.

## C COMPARISON WITH THE METHOD OPTIMIZING F-MEASURES

We also compare our method with the F1 score optimization method. Specifically, we adopt the ML$^E$ method (Ye et al., 2012) on baseline models and make comparison with our proposed COOLU method. The result is summarized in Table 6. Although method optimizing F1 score is unrelated to the Universum class, it is another way to boost model's performance. Our model also out performs the ML$^E$ method in both NER and RE tasks. In addition, the ML$^E$ method is cannot be employ on sentiment analysis task, which is evaluated by accuracy. It is another limitation of the F1 score optimization method compared to our work.

## D ABLATION STUDY ON N-PAIR LOSS PRETRAINING

In our method, N-pair loss is adopted for pretraining to learn initial representations and to speed up the training process To demonstrate the effectiveness of our closed boundary learning method and to rule out the possibility that the improvement of our model is due to the N-pair loss pretraining

Table 7: The effect of pretraining on SpanNER (Fu et al., 2021) and SCAPT (Li et al., 2021b).

| Method | F1/accuracy |
|---|---|
| SpanNER | 92.09 |
| SpanNER + N-pair pretraining | 92.05 |
| SpanNER + COOLU (N-pair pretraining) | **93.50** |
| SCAPT | 84.13% |
| SCAPT + N-pair pretraining | 84.16% |
| SCAPT + COOLU (N-pair pretraining) | **85.06**% |

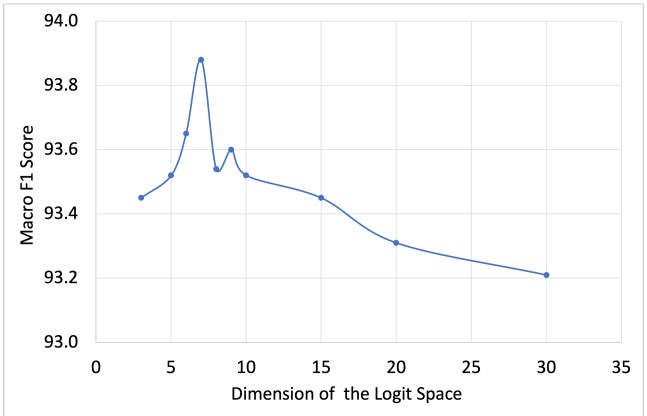

Figure 4: Illustration of generating arbitrary shape boundaries.

process, we add an additional pretraining step to baseline models of SpanNER (Fu et al., 2021) and SCAPT (Li et al., 2021b). Table 7 indicates that the pretraining process alone cannot improve the accuracy of the original baseline models. The improvement brought about by our proposed closed boundary learning is not a result of the pretraining step but the result of the entire system.

## E    THE IMPACT OF THE FINAL LAYER DIMENSION

The last layer's dimensionality can affect the performance of the model. Recalling the classic Hughes phenomenon (Hughes, 1968) that the model accuracy is monotonically increasing first and then monotonically decreasing with the dimension of data, the dimension of the final layer may be chosen to boost model performance.

We investigate the effect of last layer dimension on the accuracy of the model on the SpanNER (Fu et al., 2021) with closed boundary learning and present the result in Figure 4. The F1 score of the test set grows with increasing of dimensions and reaches a maximum value of 93.88 when the dimension is seven, and then decreases with the dimension. The trend fits well with the Hughes phenomenon Hughes (1968). Our method is quite robust with the dimension and the overall result of SpanNER + COOLU reported in Table 2 is set as ten rather than the optimal value.

