# OpenReview forum: "Closed Boundary Learning for NLP Classification Tasks with the Universum Class"
_ICLR.cc/2023/Conference — Submitted to ICLR 2023_

### Official Review · Reviewer_sUU6 · 2022-10-23

**Confidence:** 2
**Correctness:** 3
**Technical Novelty And Significance:** 2
**Empirical Novelty And Significance:** 2
**Recommendation:** 6

**Clarity, Quality, Novelty And Reproducibility:**

Clarity:
- I am unable to follow the justification in section 3.5 that arrives at the proposed p(h_k,U) in Equation (4).
- In section 3.6, the boundary learning loss is optimised over wrongly classified samples. Are these misclassified examples of the training set, or validation set? An analysis of how this loss function improves F1 while also optimising for cross-entropy could be useful to help understand the benefits of the approach.

Quality:
- the arguments are hard to follow, and there is little analysis on the results to support the arguments.

Novelty:
- the work is novel in addressing the universum boundary.

Reproducibility:
- the authors mentioned they will release the code on GitHub.




**Strength And Weaknesses:**

Strengths:
- the approach shows that it improves performance on 3 different NLP tasks, by simply learning GMM and threshold values of the positive classes.

Weaknesses:
- There is little analysis to show how the learned boundaries can better model class boundaries. A detailed analysis of precision/recall would be useful, and whether the approach improves multi-class decision boundaries, or only the boundary with the negative class.
- The proposed approach emphasises on misclassified positive samples, which could have a similar effect on work that optimises F1-measures, see e.g., [1].

Update after author rebuttal:
- The authors have addressed some of my concerns mentioned above. Hence I have increased the score to 6.

[1] Ye, N., Chai, K. M., Lee, W. S., & Chieu, H. L. (2012). Optimizing F-measures: A tale of two approaches. In Proceedings of the 29th International Conference on Machine Learning (pp. 289-296). Omnipress.

**Summary Of The Paper:**

The paper addresses the universum class (negative class) problem in NLP tasks, by modelling the class boundaries with Gaussian mixture models, and learning thresholds with a boundary learning loss on misclassified examples. They showed that their approach outperforms vanilla classifiers for 3 different NLP tasks.




**Summary Of The Review:**

I would not recommend acceptance as I think the arguments made by the authors are not supported by sufficient analysis of the results. The authors could also compare against approaches that directly optimises for f1-measures.

---

> ### Author Response · Authors · 2022-11-19
> **Response to Reviewer sUU6 (Part 1/2)**
>
> Thank you for taking the time to review our paper and providing such insightful and constructive feedback! We summarize the changes made to the paper inspired by your feedback and respond to your comments below.
>
> ---
> __[C1]:  There is little analysis to show how the learned boundaries can better model class boundaries. A detailed analysis of precision/recall would be useful.__
>
> **[A1]: Thank you for the constructive suggestion. We have analyzed precision and recall and how they reflect the learning outcome of decision boundaries in the revised paper as you suggested.**
>
> Specifically, our proposed closed boundary learning significantly improves the precision score of all classes, with the largest absolute gain being 9.03. We also observe a slight drop in recall as a result of replacing open decision boundaries with closed ones. But the increase in precision considerably outweighs the decrease in recall. The improvement in precision score indicates that the proposed closed decision boundaries of target classes can effectively avoid the misclassification of the Universum class into target classes. Simultaneously, as fewer Universum samples are misclassified, more Universum samples are correctly identified in the space outside all closed boundaries of target classes.
>
> **Revised Manuscript:** The analysis is in Section 4.3, and the results of the precision and recall are summarized in Table 3.
>
> ---
> __[C2]:  The proposed approach emphasises on misclassified positive samples, which could have a similar effect on work that optimises F1-measures. The authors could also compare against approaches that directly optimises for f1-measures.__
>
> **[A2]: Thank you for your kind suggestion. We add experiments to compare our method with the method optimizing the F1 score [1]. Our method also shows better performance.**
>
> * We adopt the ML^E method [1] proposed in the work that you kindly suggested and compare the performance of our method with the F1 score optimization method. Experimental results show that our method outperforms the ML^E method. Moreover, the methods optimizing the F1 score are not suitable for tasks evaluated by accuracy, such as the sentiment analysis task used in our paper. Although both are designed to boost the model's performance, our proposed methods are different from the F1 score optimization method as we are addressing the Universum (miscellaneous) class, which is one of the classes defined in the task but exhibits significantly different properties: heterogeneity and lack of representativeness in training data.
>
> **Revised Manuscript:** The experiments compared with the ML^E method are in Appendix C.
>
> * Inspired by your suggestion, we compare our method with one more approach. We apply OECC [2], a recent outlier exposure method for OOD, to all six SOTA baseline models for comparison with our method. Our method outperforms the OECC method. Actually, applying the OECC method results in an accuracy drop. This is because we are raising a new problem, and to our knowledge, no prior works are specially designed to address the classification task involving the Universum class. Therefore, even though the outlier exposure has the closest problem setting to ours, the OECC method is still not suitable for our problem. More detailed reasons for the accuracy drop are also analyzed in the revised manuscript.
>
> **Revised Manuscript:** We add the experiments in Table 4 and discuss the results in Section 4.2.
>
> ---
> __[C3]:  I am unable to follow the justification in section 3.5 that arrives at the proposed p(h_k,U) in Equation (4).__
>
> **[A3]: We are sorry for not addressing the justifications clearly. We rewrite the analytical part on how we arrive at our proposed p(h_k,U) in Equation (4).**
>
> Firstly, we revise the motivation for our proposed revised rule-based probability estimation for the Universum class to better clarify the general idea underlying the proposed estimation method, which is converting Rule 1 into a probability expression, while simultaneously facilitating the learning of the neural network. Then, we analyze what conditions need to be satisfied for compliance with Rule 1. We also analyze how to make the proposed estimation facilitates the learning of the neural network, and what parameters need to be involved in the function to achieve this goal. We also involve more symbols in the new version to make the analysis can be more easily connected with the proposed function.
>
> **Revised Manuscript:** The revised analysis is in Section 3.4.1

---

> > ### Author Response · Authors · 2022-11-19
> > **Response to Reviewer sUU6 (Part 2/2)**
> >
> > __[C4]: The boundary learning loss is optimised over wrongly classified samples. Are these misclassified examples of the training set, or validation set? An analysis of how this loss function improves F1 while also optimising for cross-entropy could be useful to help understand the benefits of the approach.__
> >
> > **[A4]: Thank you for the constructive suggestions. The misclassified examples are from the training set. We have also analyzed deeper into the boundary learning loss as you suggested.**
> >
> > The boundary learning loss forces misclassified samples to be distributed in the correct region, which works well with cross-entropy loss. It also guides the model to pay more attention to misclassified samples and avoids putting too much effort on correctly classified samples by cross-entropy loss. Moreover, the boundary learning loss is defined as the sum of each boundary's false positive (FP) samples and false negative (FN) samples. For instance, if there are considerably more FN samples than FP samples, the loss caused by FN samples will be larger, causing the boundary to adjust to eliminate more FN samples and balance the number of FN and FP samples. Hence, the boundary learning loss can further improve the F1 score.
> >
> > **Revised Manuscript:** The revised analysis is in Section 3.5.
> >
> > ---
> > __There are some other revisions that have been made and we wish to highlight them here.__
> >
> > **(1) Most importantly, we add additional experiments to evaluate the robustness of our model.**
> >
> > The key intention for us to do this work is to provide a more reasonable way of learning by conforming to the natural properties of the Universum class (unlike target classes with the intrinsic predictive rule, the Universum class exhibits an extrinsic predictive rule that samples are labeled as Universum if they do not belong to any of the predefined target classes.). However, we found that we failed to deliver and validate this intention in the first version of the paper. The question for us is how to prove that our method contributes to a more reasonable way of model learning. We think the method should not only be capable of improving the model's accuracy but also its robustness.
> >
> > Therefore, we evaluate the robustness of the model based on TextFlint [3], a robustness evaluation toolkit for NLP tasks. We find that the improvement in robustness is much more significant than accuracy, with the largest absolute F1 score improvement of over **8%** on the robustness evaluation dataset. We conclude that our method enhances both the accuracy and robustness of baseline models.
> >
> > **Revised Manuscript:** The experimental results are shown in Table 4, and discussed in section 4.4. The implementation details of the robustness evaluation are provided in Appendix B.3.
> >
> > ---
> > **(2) We have conducted the statistical test between the accuracy/F1 of the baseline models and our method..**
> >
> > The results of the statistical test show that the p-value is less than 0.01 for all six experiments in Table 2, indicating that the improvement brought about by our method is statistically significant.
> >
> > **Revised Manuscript:** Statistical tests are added in Table 2.
> >
> > ---
> > **(3) We add more concrete implementation details.**
> >
> > We add more implementation details, including the language models we choose in baseline models, the training process of our proposed method, which parameters are learnable, how GMM parameters and threshold values are initialized, and how to perform robustness evaluation.
> >
> > **Revised Manuscript:** Implementation details are added In Appendix B.
> >
> > ---
> > **(4) We add an experiment to quantitatively evaluate the compactness of all classes in test data.**
> >
> > The compactness of the Universum class is significantly worse than the target classes. It reflects our view that although the heterogeneous samples of the Universum class are easily mapped into a compact cluster for the training data, it is problematic for the test data. More detailed analyses are given in Appendix A.
> >
> > **Revised Manuscript:** Appendix A.
> >
> > ---
> > Please find the revised paper on this page. The revised parts have been highlighted in blue.
> >
> > ---
> > We are very thankful for your comments. We believe they have made our paper much stronger. We feel that it is a very beneficial process to learn from the comments and revise the paper in light of them.
> >
> > ---
> > **References**
> >
> > [1] Ye, Nan, et al. "Optimizing F-measures: a tale of two approaches." Proceedings of the 29th International Conference on International Conference on Machine Learning. 2012.
> >
> > [2] Papadopoulos, Aristotelis-Angelos, et al. "Outlier exposure with confidence control for out-of-distribution detection." Neurocomputing 441 (2021): 138-150.
> >
> > [3] Wang, Xiao, et al. "Textflint: Unified multilingual robustness evaluation toolkit for natural language processing." Proceedings of the 59th Annual Meeting of the Association for Computational Linguistics and the 11th International Joint Conference on Natural Language Processing: System Demonstrations. 2021.

---

> ### Author Response · Authors · 2022-12-06
> **Looking Forward to Your Reply**
>
> Dear Reviewer sUU6,
>
> We are very thankful for your insightful comments and benefit a lot from them. As the discussion stage is drawing to a close, we want to follow up to see if our responses address your concerns.
>
> We have run substantial additional experiments, made numerous revisions to the manuscript, and sent detailed responses to each of your comments.
>
> In light of your comments, we think the revision significantly improves the paper and addresses all your concerns, and we hope you agree! We really look forward to hearing your feedback!
>
> Authors of Paper 3853

---

> ### Author Response · Authors · 2022-12-10
> **A Friendly Reminder to Reviewer sUU6**
>
> Dear Reviewer sUU6,
>
> We once again appreciate your constructive comments and efforts in reviewing the paper. We have revised the paper, conducted additional experiments, and responded to each of your comments in detail. If there are further questions that need to explain, please kindly let us know.
>
> If you have no further questions, we're glad you concur that our response addressed your concerns and that the paper was improved in light of your comments. We would be very grateful if you would change the score accordingly.
>
> Authors of Paper 3853

---

> ### Author Response · Authors · 2022-12-13
> **Your feedback is very important to us!**
>
> Dear Reviewer sUU6,
>
> We truly appreciate your constructive comments and are grateful for the effort you put into reviewing our work.
>
> In our initial response, we sent detailed responses to each of your comments. We have conducted additional experiments, including the **comparison with F1 score optimization method** you suggested, **model's robustness evaluation experiments**, and **statistical test** to further illustrate the effectiveness of our method. The robustness evaluation experiments indicate that **our method improves both the accuracy and robustness of SOTA classification models**. We also incorporated in-depth analyses of experiment results, such as **analysis of precision/recall** you suggested. We have also clarified your confusion about Equation (4) in both the revised paper and our responses.
>
> We have received a response from *reviewer EeCa*, who re-evaluated our paper and increased the score by **3**. As this is the last day of the discussion phase, we would be grateful if you could let us know if our response successfully addresses your concerns. **Your feedback is really valuable to us, and we will treat it earnestly.**
>
> Finally, we would like to thank you again for your insightful comments, which have significantly improved our paper.
>
> Authors of Paper 3853

---

### Official Review · Reviewer_EeCa · 2022-10-25

**Confidence:** 4
**Correctness:** 3
**Technical Novelty And Significance:** 2
**Empirical Novelty And Significance:** 3
**Recommendation:** 6

**Clarity, Quality, Novelty And Reproducibility:**

* Clarity: The motivation is clear, but many technical details are not explained sufficiently/clearly (see cons #3 above).
* Quality: The major assumption made by the paper regarding why/how conventional treatments of the universum class could be an issue is not convincing (see cons #1 above)
* Novelty: The proposed approach is relatively new under the studied topic.

**Strength And Weaknesses:**

Pros:
* The motivation of the paper is clear. The proposed solution is simple and intuitive.
* The method is empirically shown to improve different models on several different tasks.

Cons:
* The first major concern is regarding the correctness of the motivation. While the motivation for special treatments of the universum class is clear and intuitive, it's not convincing to me whether the limitations of conventional multi-class classification as pointed out in this paper actually matter that much. Figure 1 illustrates that the decision boundaries learned by conventional classification models are "open", which makes it hard to distribute the universum class instances accurately. However, I doubt if this is really the case -- in the context of using deep neural nets (DNNs) as the feature encoder, the data representations (i.e., $\boldsymbol{h}$) and decision boundaries ($f(\boldsymbol{h})$) are jointly learned, so even if the universum class instances have quite different semantics, it should not be hard for DNNs to map them into one compact cluster in the last layer's representation space. In other words, the limitations illustrated in Figure 1 (c) could be a problem for simple models like logistic regression or SVM, but do not seem to be a challenge for DNNs which can learn arbitrary non-linear functions for data representations.
* The second major concern is the additional parameters and hyperparameters introduced by the proposed algorithm. Firstly, the GMM parameters need to be fit using target-class data -- when the amount of labeled data is small for some classes (e.g. few-shot learning or imbalanced learning), the GMM obtained may not be accurate/stable, and this limitation has not been discussed in the paper. Secondly, a threshold value ($\xi$) needs to be set for each class. It seems to be a hyperparameter that cannot be automatically learned, but the paper also didn't discuss how it can be appropriately set. The authors also didn't show how sensitive the model is to these threshold values. This is quite concerning because it's impossible to manually set threshold values for each class, especially when the number of classes is large (e.g., in fine-grained NER/RE).
* The paper is unclear in many technical details. For example, Section 3.3 mentions using N-pair loss for pretraining, but does not explain the motivation/reason. Table 4 seems to show that the pretraining step is not useful, making the purpose for pretraining very confusing. Figure 3 studies the impact of last layer dimension and reports that the optimal value is 7. This is quite confusing because the common dimensionality of pretrained models are over 700; a dimension of 7 seems way too small for the last layer dimension. Did you use another linear layer on top of the last BERT/RoBERTa encoder layer to convert the high-dimensional vector into lower-dimenisonal ones (e.g., 7-10 dimensions) before performing classification?
* The performance improvements of the proposed method over conventional classification setting are quite marginal (in most cases less than 1 point). The paper also didn't report error bars/stds or significant testing results to validate whether the improvements are meaningful.

---
**Post-Rebuttal Updates**:
I'd like to thank the authors for their provided responses and paper updates to address my concerns raised above. The original paper version had many unclear technical details which made assessing the contribution of the paper quite difficult. However, in the authors' rebuttal and updated paper version, many details have been further clarified, and I found them largely strengthened the contribution. Detailed comments are as below:
* The difficulty of learning with heterogeneous samples in the universum class has been clarified. I see that the miscellaneous patterns from the universum class that do not appear in training but in test time could be an issue for traditional classification models. However, I feel that some concrete case studies would help (e.g., some embedding plots of the actually learned classification boundaries by the proposed method); Figure 1 does not seem to be the actual plots but looks more like conceptual illustrations.
* The learning process of GMM (especially how the threshold values are learned) and its limitations (e.g., hard to adapt to few-shot learning) have been better articulated.
* The motivation for pretraining looks clearer to me.
* Glad to see the added significant tests and the new OECC baseline.

Overall, I found the paper to have studied an interesting problem (i.e., handling the universum class in classification problems), and the method proposed to be reasonable, resulting in good effectiveness. However, I feel that the application scenarios of the method are somewhat narrow (i.e., there are many classification problems in NLP that do not have universum classes at all) and the method also has inherent limitations of not being applicable to low-data regimes (e.g., few-shot learning). Although it does not appear to be a very strong and suitable paper for ICLR, I also don't want to gatekeep it. Hence, I'm recommending a weak accept.

**Summary Of The Paper:**

The paper studies how to appropriately handle the universum (or miscellaneous) class in several common NLP classification problems, such as the "non-entity" label in NER, "no-relation" label in RE. The motivation of the paper is that these universum classes can have a wide coverage of cases whose semantics vary a lot and thus should not be treated as one category as a whole. That is to say, the decision boundary of the universum class should be in different shapes from other target classes with specific semantics. The solution proposed by the paper is to learn closed boundaries for target classes, and regard the remaining points that lie out of the boundaries of any target class as the universum case. The concrete implementation is to leverage GMM to estimate the class boundaries, and use thresholds over class-conditioned generation probabilities to classify whether a data point belongs to any target class or it should be put into the universum label. The method is evaluated on three different tasks instantiated with six SOTA models and demonstrates improvements over conventional open-boundary learning.

**Summary Of The Review:**

While the paper studies an interesting issue in several NLP problems regarding how to appropriately handle the universum (or miscallaneous) class, the major motivation seems not well-justified in the context of using DNN encoder models. There are also concerns regarding the stability/generalization ability of the model due to the additionally introduced parameters and hyperparameters, and the lack of studies for these. Many technical details have not been sufficiently discussed and thus remain confusing. The empirical advantage over standard classification baseline is quite marginal.

---

> ### Author Response · Authors · 2022-11-19
> **Response to Reviewer EeCa (Part 1/3)**
>
> Thank you for taking the time to review our paper and providing such insightful and constructive feedback! We summarize the changes made to the paper inspired by your feedback and respond to your comments below.
>
> ---
> __[C1]:  The first major concern is regarding the correctness of the motivation. In the context of using deep neural nets (DNNs), even if the universum class instances have quite different semantics, it should not be hard for DNNs to map them into one compact cluster in the last layer's representation space.__
>
> **[A1]: We are sorry for not addressing this issue clearly in the paper and we we really appreciate you bringing this up. The key point to answer this issue is that such heterogeneous samples are easily mapped into a compact cluster for the training set by DNN, but it is problematic for the test set.**
>
> **From a theoretical point of view:**
>
> Why it is problematic for the test set? Recall that the Universum class is composed of highly heterogeneous samples and  Universum samples are labeled according to an extrinsic pattern: they do not belong to any of the predefined target classes. Since the natural predictive rule of the Universum class is an **extrinsic** pattern that does not belong to any target classes, the model is more likely to fit the noise in the Universum class by memorizing various peculiarities of **intrinsic** *heterogeneous* samples rather than finding the general predictive rule. Considering the data distribution of the test set differs from the training set, only memorizing various peculiarities can easily lead to **overfitting** and result in an accuracy drop. Moreover, the lack of **robustness** is another consequence of inability to find the extrinsic predictive rule for the Universum class.
>
> Another problem caused by the property of the Universum class is that classifiers with open boundaries are easily misclassifying unseen samples in the test set that are not represented by the training data.
>
> We have added a detailed and in-depth theory analysis to the revised paper to clarify the problems raised by the Universum class more clearly.
>
> **Revised Manuscript:** Theoretical analysis is added in Section 1 Introduction.
>
> **From an experimental point of view:**
>
> * We conduct additional experiments to evaluate the compactness of the Universum class and target classes in the test data. The experimental results show that the compactness of the Universum class is significantly worse than that of target classes. Considering models are more likely to overfit the noise in the Universum class rather than find the general predictive rule, it is hard for the model to map all test samples of the Universum class into the compact space learned from training. The Universum samples that deviate from the original compact space are easily misclassified, and the compactness of the Universum class in the test set is consequently worse.
>
> **Revised Manuscript:** Compactness evaluation experiments are added in Appendix A.
>
> * We evaluate the robustness of the model, and the improvement in robustness is much more significant than accuracy, with the largest absolute F1 score improvement of over 8% on the robustness evaluation dataset. The improvement in robustness reflects the correctness of our theoretical analysis above.
>
> **Revised Manuscript:** model robustness evaluation experiments are added in Section 4.4.
>
> * The analysis on the MicroF1, precision, and recall also support the theory. The precision scores of all target classes are significantly improved by adopting our method, with the largest absolute gain being 9.03. It indicates that the proposed closed decision boundary can avoid the misclassification of the Universum samples into target classes. It reflects there are many Universum samples being misclassified by DNN, i.e., deviating from the original compact space generated by training data.
>
> **Revised Manuscript:** Precision score analysis is illustrated in Section 4.3.

---

> > ### Author Response · Authors · 2022-11-19
> > **Response to Reviewer EeCa (Part2/3)**
> >
> > ---
> > __[C2]: The second major concern is the additional parameters and hyperparameters introduced by the proposed algorithm. Firstly, the GMM parameters need to be fit using target-class data. Secondly, a threshold value (ξ) needs to be set for each class.__
> >
> > **[A2]: Thank you for the constructive comments. The two points being raised are very important details and we didn't address them clearly in the first version of the paper. We have discussed the limitation of GMM estimation regarding sample size in the paper. The threshold values are automatically learned in our method. We have revised the paper to introduce it more clearly and also discuss how to initialize the threshold values in implementation details.**
> >
> > Specifically,
> >
> > * In the revised paper, we have addressed the limitation regarding the GMM estimation and the sample size. In particular, We highlight the limitation that our method is not suitable for the few-shot or zero-shot problems as the number of samples should be at least equal to the dimension of the data. Nevertheless, given that the data dimension in our method is very small (less than 20) and the initialized GMM parameters will be fine-tuned by the neural network, the sample size is not a problem for most classification problems with thousands of samples per class. Finally, for the imbalanced dataset, the class with very few samples only has a negligible effect on the overall performance, thereby diminishing the effect of inaccurate GMM estimation of the class with very few samples.
> >
> > **Revised Manuscript:** The constraint of sample size for accurate GMM estimation is addressed in Section 3.3.1.
> >
> > * Regarding the question of how to set the threshold values, we are sorry for not addressing this issue clearly in the paper. The threshold values are different for each target class and they are learnable parameters in our method. Specifically, the threshold values are learned based on the balance of misclassified samples inside and outside the boundary by our proposed boundary learning loss. We also introduce how to initialize the thresholds in Appendix B Implementation Details of the revised paper.
> >
> > **Revised Manuscript:** How to learn the threshold values is discussed in Section 3.3.2, and the initialization of threshold values is introduced in Appendix B.2.
> >
> > ---
> > __[C3]: The paper is unclear in many technical details, including not explaining the motivation of the pretraining and the dimension of the last layer__
> >
> > **[A3]: Thank you for the constructive comments. We have made clarifications on the two points raised in the comment in the revised paper. We also add more implementation details in the Appendix.**
> >
> > Specifically,
> >
> > * Motivation for pertaining: Our method estimates the probability distribution of target classes based on their sample distributions. In order to avoid estimation based on randomly initialized weight and speed up the learning process, we employ N-pair loss for pretraining. And the purpose of the ablation study on pretraining is to demonstrate the effectiveness of our closed boundary learning method and to rule out the possibility that the improvement of our model is due to the N-pair loss pretraining process.
> >
> > **Revised Manuscript:** The motivation for pretraining is illustrated in Section 3.2, and the purpose of the ablation study on pretraining is discussed in Appendix D.
> >
> > * Clarification of the final layer: To make our method compatible with general classification methods, the starting point of our method is the final layer of classification models, which is a linear layer that maps data from a high-dimensional feature space to a lower-dimension space. And classification is performed on the linear layer's output.
> >
> > **Revised Manuscript:** The clarification of the final layer is provided at the beginning of Section 3.
> >
> > * We also clarify how the proposed boundary learning loss works with the cross-entropy loss, and illustrate the ability of boundary learning loss to improve the F1 score.
> >
> > **Revised Manuscript:** Details of boundary learning loss are discussed in Section 3.5.
> >
> > * We add more implementation details, including the language models we choose in baseline models, the training process of our proposed method, which parameters are learnable, how GMM parameters and threshold values are initialized, and how to perform robustness evaluation.
> >
> > **Revised Manuscript:** Implementation details are added in Appendix B.

---

> > > ### Author Response · Authors · 2022-11-19
> > > **Response to Reviewer EeCa (Part 3/3)**
> > >
> > > ---
> > > __[C4]:  The improvements of the proposed method over conventional classification setting are quite marginal. The paper also didn't report error bars/stds or significant testing results__
> > >
> > > **[A4]: Thank you for the constructive comments. As you suggested, we have conducted the statistical test between the accuracy/F1 of the baseline models and our method. Experimental results show the improvements are statistically significant. In addition, we conduct additional experiments on the robustness evaluation. Our method not only improves the accuracy of baseline models but also significantly enhances the robustness of the model**
> > >
> > > * The results of the statistical test show that the p-value is less than 0.01 for all six cases, indicating that the improvement brought about by our method is statistically significant.
> > >
> > > **Revised Manuscript:** Statistical tests are added in Table 2.
> > >
> > > * We evaluate the robustness of the model based on TextFlint [1], a robustness evaluation toolkit for NLP tasks. We find that the improvement in robustness is much more significant than accuracy, with the largest absolute F1 score improvement of over 8 on the robustness evaluation dataset. The detailed experiments and analysis are added to the revised paper.
> > >
> > > **Revised Manuscript:** The results are shown in Table 4, and discussed in section 4.4. The implementation details of the robustness evaluation are provided in Appendix B.3.
> > >
> > > ---
> > > __There are some other revisions that have been made and we wish to highlight them here.__
> > >
> > > **(1) We add additional experiments to compare our method with other works in dealing with the classification tasks involving the Universum class**
> > >
> > > We apply OECC [2], a recent outlier exposure method for OOD, to all six SOTA baseline models for comparison with our method. Our method outperforms the OECC method. Actually, applying the OECC method results in an accuracy drop. This is because we are raising a new problem, and to our knowledge, no prior works are specially designed to address the classification task involving the Universum class. Therefore, even though the outlier exposure has the closest problem setting to ours, the OECC method is still not suitable for our problem. More detailed reasons for the accuracy drop are also analyzed in the revised manuscript.
> > >
> > > **Revised Manuscript:** We add the experiments in Table 4 and discuss the results in section 4.2.
> > >
> > > **(2) We reconstruct the motivation of our proposed extrinsic rule-based probability estimation for the Universum Class.**
> > >
> > > Intuitively dealing with the Universum class by adopting Universum detection and multi-class classification in two steps will result in the problem of error propagation. The challenge for jointly learning the Universum class and target classes is how to estimate the probability distribution of the Universum class without relying on intrinsic sample distribution. Our proposed strategy converts Rule 1 (A sample is assigned to the Universum class if it is not located inside any of the closed boundaries of target classes) into a probability expression, while simultaneously facilitates the learning of the neural network. The extrinsic rule-based probability estimation jointly learns the Universum class and target class and avoids the error propagation of the intuitive two-step method.
> > >
> > > **Revised Manuscript:** Section 3.4.
> > >
> > > **(3) We provide a more reasonable way of learning by conforming to the natural properties of the Universum class.**
> > >
> > > The key intention for us to do this work is to provide a more reasonable way of learning by conforming to the natural properties of the Universum class. However, we found that we failed to deliver and validate this intention in the first version of the paper. The question for us is how to prove that our method contributes to a more reasonable model learning process. Inspired by your comments, we find we didn't express our motivation clearly enough. And we also find the results on accuracy improvements are not enough to validate our intention. It drives us to make modifications in the introduction and conduct additional experiments on robustness evaluation.
> > >
> > > **Revised Manuscript:** In Section 1 and Section 4.4.
> > >
> > > ---
> > > Please find the revised paper on this page. The revised parts have been highlighted in blue
> > >
> > > ---
> > > We are very thankful for your comments. We believe they have made our paper much stronger. We feel that it is a very beneficial process to learn from the comments and revise the paper in light of them.
> > >
> > > **References**
> > >
> > > [1] Wang, Xiao, et al. "Textflint: Unified multilingual robustness evaluation toolkit for natural language processing." Proceedings of the 59th Annual Meeting of the Association for Computational Linguistics and the 11th International Joint Conference on Natural Language Processing: System Demonstrations. 2021
> > >
> > > [2] Papadopoulos, Aristotelis-Angelos, et al. "Outlier exposure with confidence control for out-of-distribution detection." Neurocomputing 441 (2021): 138-150.

---

> ### Author Response · Authors · 2022-12-06
> **Looking Forward to Your Reply**
>
> Dear Reviewer EeCa,
>
> We are very thankful for your insightful comments and benefit a lot from them. As the discussion stage is drawing to a close, we want to follow up to see if our responses address your concerns.
>
> We have run substantial additional experiments, made numerous revisions to the manuscript, and sent detailed responses to each of your comments.
>
> In light of your comments, we think the revision significantly improves the paper and addresses all your concerns, and we hope you agree! We really look forward to hearing your feedback!
>
> Authors of Paper 3853

---

> ### Author Response · Authors · 2022-12-10
> **A Friendly Reminder to Reviewer EeCa**
>
> Dear Reviewer EeCa,
>
> We once again appreciate your constructive comments and efforts in reviewing the paper. We have revised the paper, conducted additional experiments, and responded to each of your comments in detail. We specifically provide both theoretical and experimental analysis in response to your major concern about the motivation of this work. If there are further questions that need to explain, please kindly let us know.
>
> If you have no further questions, we're glad you concur that our response addressed your concerns and that the paper was improved in light of your comments. We would be very grateful if you would change the score accordingly.
>
> Authors of Paper 3853

---

> ### Comment · Reviewer_EeCa · 2022-12-13
> **Post-Rebuttal Updates**
>
> I'd like to thank the authors for their provided responses and paper updates to address my concerns. I believe that the authors' rebuttal and updated paper version have successfully clarified many details, and I found them largely strengthened the contribution of the paper. Please refer to **Post-Rebuttal Updates** in my main review above for detailed comments. I've increased my rating from 3 to 6.

---

> > ### Author Response · Authors · 2022-12-13
> > **Thank you for your response and your careful reading of our rebuttal**
> >
> > Dear Reviewer EeCa,
> >
> > Thank you very much for your continuous effort in reviewing the paper in the rebuttal phase and for all your constructive comments! We really appreciate your being willing to increase the score.
> >
> > We have the following responses to your comments in the post-rebuttal updates.
> >
> > * We are very happy to learn that our response could help to address your concerns well.
> >
> > * Your suggestion on concrete case studies of embedding plots to illustrate the miscellaneous pattern is very helpful and we will incorporate them into the final version of the paper.
> >
> > * In terms of the application scenarios, we find the Universum class widely exists in many NLP tasks, including the no relation class in **relation extraction**, the miscellaneous class in **named entity recognition**, the neutral class in **sentiment analysis**, the other class in **event extraction**, the neutral class in **natural language inference**, the other class in **extractive summarization**, etc. In addition, we think our method is potentially useful in real-life applications. Compared with generally used open boundary classifiers, our method is capable to learn the **model's knowledge boundary** (closed boundary) of each target class. The space inside the boundary represents what the model *knows* about a certain class, i.e., the recognized patterns, after learning from training samples, whereas the space outside the boundary represents what the model *doesn’t know* about this class from training data. Such knowledge boundary is useful in safety-critical applications such as autonomous vehicles. How to leverage the model's knowledge boundary learned by our method can be an interesting question for our future works. We hope the above explanations can help alleviate your concerns about application scenarios.
> >
> > Finally, we would like to thank you again not only for your support on our paper and careful reading of our rebuttal, but also for your insightful comments throughout the entire review process, which have significantly improved our paper.
> >
> > Sincerely,
> >
> > Authors of Paper 3853

---

### Official Review · Reviewer_CgAh · 2022-10-26

**Confidence:** 3
**Correctness:** 3
**Technical Novelty And Significance:** 3
**Empirical Novelty And Significance:** 2
**Recommendation:** 6

**Clarity, Quality, Novelty And Reproducibility:**

Quite good presentation of the material and as well as the motivation. There is some novelty in the loss function and the authors provide enough details even though I would encourage them to add more material regarding the implementation/setting in the Appendix.

**Strength And Weaknesses:**

- The paper is easy to follow and intuitive. Explanations of the different parts is adequate.
- The authors present several related works also and do some comparisons.
- Results seem to be ok even though only on three datasets.

Regarding weakness one of the first comments is the lack of other approaches that treat the same problem. The authors cite several related works in related cases (OOD detections, anomalies etc) but do not do any comparisons. I mean, the fact that they enhance few SOTA approaches is already good but I cannot assess how this pushes the current state-of-the-art methods in the specific problem.

Lack of statistical tests to evaluate significance. In some cases improvements are quite small. Maybe you could add them?

I am curious if any other baseline like for example to consider the universum class as a side task would work here.

**Summary Of The Paper:**

The authors propose a method to learn closed boundaries on top of the final layer of a classification model. The authors propose to optimize the classifier with an appropriate loss adjusted by the miss-classified examples within and outside the boundaries. The authors, extend state-of-the-art approaches in three datasets. The results show some improvement in all the cases.

**Summary Of The Review:**

Generally the paper is easy to read and flows well. I have some difficulty to assess the improvements that this approach brings. Based on the presented results the proposed approach seems to help improving the models that it extends.

---

> ### Author Response · Authors · 2022-11-19
> **Response to Reviewer CgAh (Part 1/2)**
>
> Thank you for taking the time to review our paper and providing such insightful and constructive feedback! We summarize the changes made to the paper inspired by your feedback and respond to your comments below.
>
> ---
> __[C1]:  The lack of other approaches that treat the same problem.__
>
> **[A1]: Thank you for pointing out the problem. We have conducted additional experiments on this issue.**
>
> Specifically,
>
> * We apply OECC [1], a recent outlier exposure method for OOD, to all six SOTA baseline models for comparison with our method. Our method outperforms the OECC method. Actually, applying the OECC method results in an accuracy drop. This is because we are raising a new problem, and to our knowledge, no prior works are specially designed to address the classification task involving the Universum class. Therefore, even though the outlier exposure has the closest problem setting to ours, the OECC method is still not suitable for our problem. More detailed reasons for the accuracy drop are also analyzed in the revised manuscript.
>
> **Revised Manuscript:** We add the experiments in Table 4 and discuss the results in Section 4.2.
>
> * We add experiments to compare our method with the method optimizing the F1 score [2]. Our method also shows better performance.
>
> **Revised Manuscript:** We add the experiments in Appendix C.
>
> ---
> __[C2]:  Lack of statistical tests to evaluate significance.__
>
> **[A2]: Thank you for the constructive suggestion. We have conducted the statistical test between the accuracy/F1 of the baseline models and our method.**
>
> The p-value is less than 0.01 for all six cases, indicating that the improvement brought about by our COOLU method is statistically significant.
>
> **Revised Manuscript:** Statistical tests are added in Table 2.
>
> ---
> __[C3]: I am curious if any other baseline like for example to consider the universum class as a side task would work here.__
>
> **[A3]: Thank you for the interesting idea. The question is analyzed below**
>
> If we understood correctly, considering the Universum class as a side task is conducting Universum detection alongside multi-class classification. This is a very good idea as it is the structure for many outlier exposure methods. The OECC method [1] we added to the experiments also follows this structure. However, it may fail in our problem because it will suffer from the error propagation problem by classifying in two steps: multi-class class classification is based on the Universum detection results. That's why we propose the extrinsic rule-based probability estimation for the Universum class in section 3.4 to jointly classify the Universum class and target classes.
>
> **Revised Manuscript:** It would be similar to the OECC results in Table 4.
>
> ---
> __[C4]: I would encourage them to add more material regarding the implementation/setting in the Appendix.__
>
> **[A4]: Thank you for the kind suggestion. We have added much more implementation details in the Appendix**
>
> The implementation details have been added to Appendix B, including B.1 baseline models, B.2 training process, and B.3 robustness evaluation. Specifically, the process of training, which parameters are learnable, and how GMM parameters and threshold values are initialized are summarized in Appendix B.2.
>
> **Revised Manuscript:** Implementation details are added in Appendix B.

---

> > ### Author Response · Authors · 2022-11-19
> > **Response to Reviewer CgAh (Part 2/2)**
> >
> > __There are some other revisions that have been made and we wish to highlight them here.__
> >
> > ---
> > **(1) Most importantly, we add additional experiments to evaluate the robustness of our model.**
> >
> > The key intention for us to do this work is to provide a more reasonable way of learning by conforming to the natural properties of the Universum class. However, we found that we failed to deliver and validate this intention in the first version of the paper. The question for us is how to prove that our method contributes to a more reasonable way of model learning. We think the method should not only be capable of improving the model's accuracy but also its robustness.
> >
> > Therefore, we evaluate the robustness of the model based on TextFlint [3], a robustness evaluation toolkit for NLP tasks. We find that the improvement in robustness is much more significant than accuracy, with the largest absolute F1 score improvement of over 8% on the robustness evaluation dataset. The detailed experiments and analysis are added to the revised paper.
> >
> > **Revised Manuscript:** The results are shown in Table 4, and discussed in section 4.4. The implementation details of the robustness evaluation are provided in Appendix B.3.
> >
> > ---
> > **(2) We add more detailed and in-depth analysis to some parts of the paper.**
> >
> > * We further analyze the consequence of classifying the Universum class with open decision boundaries.
> >
> > **Revised Manuscript:** In the introduction part.
> >
> > * We reconstruct the motivation of our proposed extrinsic rule-based probability estimation for the Universum Class. Our proposed strategy converts Rule 1 (A sample is assigned to the Universum class if it is not located inside any of the closed boundaries of target classes) into a probability expression, while simultaneously facilitates the learning of the neural network. The extrinsic rule-based probability estimation achieves jointly learning the Universum class and target class joint and avoids the error propagation of the intuitive two-step method.
> >
> > **Revised Manuscript:** Section 3.4.
> >
> > * We further analyze how boundary learning loss can contribute to boundary learning and better F1 score.
> >
> > **Revised Manuscript:** Section 3.5.
> >
> > ---
> > **(3) We add an experiment to quantitatively evaluate the compactness of all classes in test data.**
> >
> > The compactness of the Universum class is significantly worse than the target classes. It reflects our view that although the heterogeneous samples of the Universum class are easily mapped into a compact cluster for the training data, it is problematic for the test data. More detailed analyses are given in Appendix A.
> >
> > **Revised Manuscript:** Appendix A.
> >
> > ---
> > Please find the revised paper on this page. The revised parts have been highlighted in blue.
> >
> > ---
> > We are very thankful for your comments. We believe they have made our paper much stronger. We feel that it is a very beneficial process to learn from the comments and revise the paper in light of them.
> >
> > ---
> > **References**
> >
> > [1] Papadopoulos, Aristotelis-Angelos, et al. "Outlier exposure with confidence control for out-of-distribution detection." Neurocomputing 441 (2021): 138-150.
> >
> > [2] Ye, Nan, et al. "Optimizing F-measures: a tale of two approaches." Proceedings of the 29th International Conference on International Conference on Machine Learning. 2012.
> >
> > [3] Wang, Xiao, et al. "Textflint: Unified multilingual robustness evaluation toolkit for natural language processing." Proceedings of the 59th Annual Meeting of the Association for Computational Linguistics and the 11th International Joint Conference on Natural Language Processing: System Demonstrations. 2021.

---

> ### Author Response · Authors · 2022-12-06
> **Looking Forward to Your Reply**
>
> Dear Reviewer CgAh,
>
> We are very thankful for your insightful comments and benefit a lot from them. As the discussion stage is drawing to a close, we want to follow up to see if our responses address your concerns.
>
> We have run substantial additional experiments, made numerous revisions to the manuscript, and sent detailed responses to each of your comments.
>
> In light of your comments, we think the revision significantly improves the paper and addresses all your concerns, and we hope you agree! We really look forward to hearing your feedback!
>
> Authors of Paper 3853

---

> ### Author Response · Authors · 2022-12-10
> **A Friendly Reminder to Reviewer CgAh**
>
> Dear Reviewer CgAh,
>
> We once again appreciate your constructive comments and efforts in reviewing the paper. We have revised the paper, conducted additional experiments, and responded to each of your comments in detail. If there are further questions that need us to explain, please kindly let us know.
>
> If you have no further questions, we're glad you concur that our response addressed your concerns and that the paper was improved in light of your comments. We would be very grateful if you would change the score accordingly.
>
> Authors of Paper 3853

---

> ### Author Response · Authors · 2022-12-13
> **Your feedback is very important to us!**
>
> Dear Reviewer CgAh,
>
> Thank you so much for your support of our paper. We truly appreciate your constructive comments and are grateful for the effort you put into reviewing our work!
>
> In our initial response, we sent detailed responses to each of your comments. We have conducted substantial additional experiments, including the **comparison with OOD detection method** and **statistical test** as you suggested. The experimental results indicate that our model outperforms OOD detection approaches and our improvements are statistically significant. We also conducted additional **model's robustness evaluation experiments** to further demonstrate the effectiveness of our method. The robustness evaluation experiments indicate that **our method improves both the accuracy and robustness of SOTA classification models**. In addition, **many more implementation details** are included in the appendix following your suggestion.
>
> We have received a response from *reviewer EeCa*, who re-evaluated our paper and increased the score by **3**. As this is the last day of the discussion phase, we would be grateful if you could let us know if our response successfully addresses your concerns. **Your feedback is really valuable to us, and we will treat it earnestly**.
>
> Finally, we would like to thank you again for your insightful comments, which have significantly improved our paper.
>
> Authors of Paper 3853

---

### Decision · Program_Chairs · 2023-01-20

**Decision:**

Reject

**Justification For Why Not Higher Score:**

Basically, the paper needs another round of work to improve contextualization wrt existing work, discussion of empirical results, tying up a few loose ends, and overall integration of these results into the narrative.

**Justification For Why Not Lower Score:**

N/A

**Metareview: Summary, Strengths And Weaknesses:**

To specifically address the universum class for NLP tasks (i.e., the outside/rejection/miscellaneous class), the authors propose to learn closed boundaries for the known classes on top of the final layer of an (arbitrary) classification model by optimizing the decision boundary with a GMM-based solution for each class with a targeted misclassification loss. Based on the resulting model, the authors improve over SotA results on the six datasets by explicitly mitigating the standard open-boundary assumption.

The consensus strengths regarding this work include:
- The universum class is widely used in NLP tasks (and many general ML tasks), but hasn’t received significant explicit attention in these contexts. Thus, the authors are addressing an understudied problem.
- The proposed method is intuitive and can be easily applied to many settings to achieve a ‘free’ empirical gain in many cases.
- The paper is well-written overall (and the paper significantly improved during the rebuttal).
- The authors provided significant new results during the rebuttal period to strengthen the paper and address open issues.

The reviewer (and my own) identified weaknesses for this work include:
- As the authors are aware, closed boundary learning approaches appear in many ML settings (e.g., OOD, anomaly detection, outlier detection). It also appears in class imbalance settings (and other cases). It would strengthen the paper to survey these some and make a clear contrast to the proposed work. For example, another approach is to perform k-NN on the universum class and treat this as the closed set (assuming mixtures of spherical-based distributions) — and I expect some such ideas would surface in such a survey that may introduce more closely comparable baselines. Specifically, section 2 should be more concise, precise, and complete.
- Some of these existing works in other setting include analytical analysis that seems translatable to this problem to help with a theoretical analysis. As it stands, the paper is largely empirical and makes some claims that the reviewers didn’t feel were well-supported (i.e., the motivation is in 2D spaces).
- There remain open questions around some technical details and sensitivity analysis of hyperparameters.
- While the empirical gains are ‘free’ in that a practitioner can apply the approach without requiring significant problem specialization, they aren’t universally notable (even if they are statistically significant).
- The new results improve the technical strength of the paper significantly, but the interpretation of the overall empirical results is more disjoint now as the new results should be better integrated.

Overall, the reviewers agree that this is an understudied problem (at least within the NLP community) and there is an opportunity to modify existing algorithms to get consistent improvements with minimal effort — thus, potentially impactful. Additionally, the reviewers appreciated how much better the paper became during the rebuttal period. However, there remain concerns regarding contextualizing this work within non-NLP techniques for this problem (including possible theoretical analysis) and more interpretation/discussion of all the empirical results in the paper.

Additional note: This submission was also discussed with the SAC during final decisions. My assessment is that the original submission was a confident reject and the improvements during the rebuttal period make it a borderline case. The technical content is now strong enough to warrant consideration for acceptance, but the analysis/interpretation/contextualization would need to be improved to make it a clear accept. The SAC, program chair, and myself agreed upon this decision after reading all of the reviews, rebuttals, and discussion -- and then discussing amongst ourselves.